# Synthesis of Alginate Nanogels with Polyvalent 3D Transition Metal Cations: Applications in Urease Immobilization

**DOI:** 10.3390/polym14071277

**Published:** 2022-03-22

**Authors:** Abhishek Saxena, Shivani Sharda, Sumit Kumar, Benu Kumar, Sheetal Shirodkar, Praveen Dahiya, Rachana Sahney

**Affiliations:** 1Amity Institute of Biotechnology, Amity University Uttar Pradesh, Noida 201303, India; as0603@gmail.com (A.S.); ssharda@amity.edu (S.S.); bkumar5@amity.edu (B.K.); sshirodkar@amity.edu (S.S.); pdahiya@amity.edu (P.D.); 2Radioanalytical Chemistry Division, Radiological Laboratories, Bhabha Atomic Research Centre, Mumbai 40008, India; sumitk@barc.gov.in

**Keywords:** sodium alginate, nanogels, template-based synthesis, enzyme, polycation cross-linker

## Abstract

Biocompatible nanogels are highly in demand and have the potential to be used in various applications, e.g., for the encapsulation of sensitive biomacromolecules. In the present study, we have developed water-in-oil microemulsions of sodium alginate sol/hexane/Span 20 as a template for controlled synthesis of alginate nanogels, cross-linked with 3d transition metal cations (Mn^2+^, Fe^3+^, and Co^2+^). The results suggest that the stable template of 110 nm dimensions can be obtained by microemulsion technique using Span 20 at concentrations of 10mM and above, showing a zeta potential of −57.3 mV. A comparison of the effects of the cross-links on the morphology, surface charge, protein (urease enzyme) encapsulation properties, and stability of the resulting nanogels were studied. Alginate nanogels, cross-linked with Mn^2+^, Fe^3+^, or Co^2+^ did not show any gradation in the hydrodynamic diameter. The shape of alginate nanogels, cross-linked with Mn^2+^ or Co^2+^, were spherical; whereas, nanogels cross-linked with Fe^3+^ (Fe–alginate) were non-spherical and rice-shaped. The zeta potential, enzyme loading efficiency, and enzyme activity of Fe–alginate was the highest among all the nanogels studied. It was found that the morphology of particles influenced the percent immobilization, loading capacity, and loading efficiency of encapsulated enzymes. These particles are promising candidates for biosensing and efficient drug delivery due to their relatively high loading capacity, biocompatibility, easy fabrication, and easy handling.

## 1. Introduction

The linear copolymer of α-L-guluronic acid (G) and β-D-mannuronic acid (M) residues have an inherent ability to form alginate hydrogels by ionotropic gelation with divalent and trivalent metal cations [1]. The cation cross-linked alginate hydrogels have found applications in a variety of fields such as environmental sciences, food sciences, biomedical sciences, and nanotechnology, including use as cell scaffolds for islet cell immobilization [2,3], 3D bioprinting [4], delivery systems for pharmaceuticals [5,6,7], sorbents for the sequestration of metals in contaminated aqueous solutions [8,9], substrates for microfluidics [10], components in the production of foods and beverages [11], and food coating material for fresh fruits and cut fruits and vegetables to reduce lipid oxidation [12]. The discreet applications of the alginate hydrogels necessitate deriving the size range and shape of these gel particles, which can show superior functions in the biosystems. A vast majority of studies related to alginate gelation, structure, and application are based on alginate gel particles measurements ranging between >1 mm (macro), 0.2–1000 µm (micro), and <0.2 µm (nano), containing calcium ions as the most frequently used cross-linking agent [13,14]. The structural analysis has revealed that divalent and trivalent cations are able to form a complex with the G-units of two alginate linear polymer chains, forming a cooperative egg-box structure; thereby, two linear chains of polymer are linked by a cross-link [15,16,17]. The naturally occurring alginate’s affinity for divalent cations differ in the following order: Pb > Cu > Cd >Ba > Sr > Ca > Co, Ni, Zn > Mn [18,19]. This evokes the pre-eminent importance for the choice of ions in the preparation and application of alginate gel particles [20,21]. The insight into the chain folding mechanism of alginates and the local structures of alginate gels, developed by the interactions between uronic units and cross-linking ions, have been provided by computational studies [17,22,23]. Varied metal cations (Cu^2+^, Zn^2+^, Sr^2+^, etc.), other than Ca^2+^, are increasingly explored in novel production techniques or to tailor the properties of alginates in the development of new functional biomaterials. Studies have revealed the application of calcium ion cross-linked alginate (Ca–alginate) capsules [24] for encapsulation of magnetic iron oxide nanoparticles and its application as contrast agents in magnetic resonance imaging (MRI). Later, Morch et al. [21] showed that alginate gels cross-linked with manganese cations (Mn–alginate) with lower stability can be exploited as a system for controlled release of manganese ions (Mn^2+^) applicable in the manganese-enhanced MRI (MEMRI). These reports indicate that the mechanical properties of alginate hydrogels can be modulated with different divalent cations as needed. Furthermore, magnetically responsive hydrogels spheres can also be fabricated by cross-linking with paramagnetic ions-Ho^3+^ which assemble in magnetic field gradients. Winkleman et al. showed that it can be used as a matrix to introduce recoverable sensors or reagents to aqueous mixtures [25]. The creation of cross-linking with the available pool of cations has been realized to provide the starting material of investigations into their potential role in biological estimations. Calcium ion cross-linked alginate (Ca–alginate) is known to be an unfavorable substrate for cell-cultures; whereas, Fe–alginate is able to overcome the deficiencies of Ca–alginate, such as poor protein adsorptive capacity. Thus, Fe–alginate is an effective alternate as cell culture media [26]. As an alternate to Ca^2+^, both Ba^2+^ and Sr^2+^ produce stronger, yet still biocompatible, alginate gels [20,27,28,29]. The continued investigations into the photo-induced transition of Fe (III)/ Fe (II) alginates by Giammanco et al. [30] have shown that visible-light-responsive alginate gel beads, cross-linked with Fe (III) (Fe–alginate), can act as stable carriers for different molecules as diverse as the dye Congo Red, the vitamin folic acid, and the antibiotic chloramphenicol. The requisite abstraction and control of the Fe–alginate beads are relayed through their photo-responsive nature, which have been exploited for the controlled-release of the encapsulated cargo [30]. Microcapsules of alginate biopolymer cross-linked with Ni^2+^ and Co^2+^ have been exploited as templates for the controlled growth of Cobalt (Co) and Nickel (Ni) magnetic nanoparticles and Co–Ni nanoalloys [31]. Similarly, sodium alginate polymer with high guluronate content was ion-exchanged with transition-metal ions for the development of heterogeneous catalysts in the industrial application [32]. Iron binding properties of alginates are routinely used in the fabrication of iron–oxide nanoparticles [33]. In acidic pH conditions comparable to gastric juices, alginate and iron interact to form iron–oxide centered nanoparticles [34]. Alginate microbeads, made of high-G alginate gelled with a combination of calcium and a low concentration of barium ions, is recommended for islet transplantation in mice [20,35]. Alginate gel beads with a combination of Ca^2+^ and Co^2+^ can be used for the encapsulation of human adipose-derived mesenchymal stem cells and their application in cartilage repair [36]. Thus, unique functional properties can be added into the alginate hydrogel polymer network by incorporating the specific metal ions.

A vast majority of alginate particles exist as spheres or beads, synthesized by either internal or external gelation [16,37,38]. However, the formation of alginate nanogels is less common in comparison with nanoparticles of other synthetic polymers [14]. Nanoparticles smaller than 1 μm have many advantages over larger alginate particles in having high surface/volume ratio, high reactivity, and catalytic activity, and so on [39,40]. These properties enable higher encapsulation efficiency of biomolecules in the 3D architecture of hydrogels, lower inactivation of the bioactive molecule due to lower molecular orientation problems, and a wet environment, which improves biomolecular stability [41,42]. The preparation method used to obtain nanoparticles defines the nano-sized system as nano-aggregate [43], nano-capsules [44,45], and nano-spheres [46,47]. This calcium-based nanogel particle can be synthesized by adjusting the polymer concentration and calcium ion concentration [48]. It can be used for encapsulation of ferrous ions, which could be exploited for oral delivery of iron [49]. Xue et al. have shown that the pH responsive property of anionic sodium alginate (SA) can be used for the encapsulation of cationic doxorubicin (DOX) via electrostatic interactions, followed by in situ cross-linking with calcium ions under ultrasonic bath. These pH-responsive nanogels can be further used for cancer therapy [50]. Typically, alginate nanogels cross-linked with calcium ions are expended for various preparation methods and applications [46,50,51,52,53,54]. As an alternative to calcium, we reported the synthesis of barium–alginate and strontium–alginate nanogels [55]. The comparison of the physicochemical characteristics and enzyme encapsulation abilities of these nanogels revealed the superior mechanical and biocompatible nature of barium–alginate nanogel.

In the present study, we used reverse micellar aggregates of lipophilic surfactant as a template for alginate nanogel synthesis. The critical concentration of surfactant required for the generation of a stable reverse-micellar template for alginate nanogel is not documented. We report the inceptive details of determining the critical micelle concentration (CMC) of surfactant in the alginate sol/hexane-system required for the templating of nanogel synthesis. The template has been used for the development of alginate nanogel cross-linked with different 3d transition metal cations. Urea has always been an analyte that has received much attention because of its vital effect in clinical applications [56], food analyses, enzymatic micromotors and nanomotors [57]. The urease–urea reaction is extensively used in synthesizing photonic devices, biomaterials, sensors, and actuators, etc., all composed of polymeric materials [58]. In these immobilized, urease enzyme-based systems, the critical issue is maintaining the stability, activity, and function of the enzyme as close as possible to its native state [59]. The application of immobilized enzymes is preferred due to their ease of handling, prolonged availability, robustness, increased resistance to environmental changes, and reusability [60]. The characteristics of immobilized enzymes are controlled by the properties of both the enzyme and the support matrix. The present study is designed to prepare alginate nanogels using various polyvalent ions (Mn^2+^, Fe^3+^, and Co^2+^) as cross-links, and to compare the effects of these cross-links on the morphology, surface charge, protein (urease enzyme) encapsulation properties, and stability of the resulting nanogels. The success of urease immobilization was determined by the study of various immobilization parameters, i.e., enzyme loading and specific enzyme activity. Our study suggests new directions for the development of alginate nanogels cross-linked with transition metal cations (Mn^2+^, Fe^3+^, and Co^2+^) for different applications, including food and beverages, biosensors, wastewater treatment, and dialysate regeneration, etc. 

## 2. Materials and Methods

Sodium alginate (mol. wt. 120,000–190,000), urease, and sorbitan monolaurate (Span 20) were purchased from Sigma-Aldrich, New Delhi, India. Calcium chloride, manganous chloride, ferrous chloride, cobalt chloride, sodium chloride, Nessler’s reagent, and urea were supplied by Fisher Scientific, New Delhi, India. Hexane was supplied by CDH, New Delhi, India. All chemicals were of analytical grade (AR) and were used as received. Triple distilled water was used throughout the experiments.

### 2.1. Preparation of Alginate Nanogels and Encapsulation of Urease Enzyme

The alginate nanogels, cross-linked with different transition metal cations, were prepared as reported earlier (55). In brief, the organic phase (hexane, 10 mL), containing Span 20, was taken in a vial, and the sodium alginate solution (0.2%) was prepared in Tris-acetate-saline buffer (100 mM, pH 7.2) was added dropwise under constant stirring at 500 rotations per minute (rpm). Different phases appeared (transparent to turbid) during the process of adding alginate solution at different surfactant concentrations, which lead to the formation of water-in-oil reverse micelles (W/O). The reverse micelles so formed was characterized using dynamic light scattering (DLS) instrument and the size of microemulsion droplets were determined. The alginate nanogel particles were prepared by using W/O reverse micelle with droplet size 110 nm. The W/O reverse micelle was stirred for 30 min and 10 mL (100 mM) of filtered solutions containing divalent cations (MnCl_2,_ FeCl_3,_ CoCl_2_) were added dropwise, which led to cross-linking of the polymer chains. After careful washing with deionized water to remove surfactant and hexane, the vial was centrifuged at 3000× *g* for 30 min, and small white pellets of alginate nanogels were obtained. The pellets were re-suspended in Tris-acetate-saline buffer for successive studies. The overall procedure for the encapsulation of urease into the alginate nanogels was same as described above, except that the urease enzyme was pre-mixed with 0.2% alginate solution.

### 2.2. Characterization

#### 2.2.1. Size Determination and Surface Charge Studies

The microemulsion or the colloidal suspension of alginate nanogels, or both (1 mL sample), was placed in a quartz cuvette and analyzed using DLS (Malvern Zetasizer-Nano ZS (Malvern, UK) instrument, S-90) to estimate the mean hydrodynamic diameter and the polydispersity index (PDI). The mean hydrodynamic radii and intensity-averaged size distributions were obtained from the raw data using the general-purpose inverse Laplace transformation method provided in the instrument software. The PDIs were estimated from cumulant analysis, which is also provided with the instrument software.

To measure the zeta potential in the electrophoretic light scattering (ELS) mode, the alginate nanogels, containing different cross-link ions (Mn^2+^, Fe^3+^, Co^2+^) with 0.5 *w/v* particle concentration, were suspended in water and placed in a standard cell slowly to avoid air bubbles. When the cell was inserted into the Zetasizer, electrodes positioned on either side of the cell holder supplied the voltage necessary to perform electrophoresis. The zeta potentials were calculated automatically at 25 °C by the instrument, determining the electrophoretic mobility using the Henry equation [61]. Each sample was run in triplicate, and the average of the three readings was reported.

#### 2.2.2. SEM-EDX Measurement

The surface morphologies (shape and formation of aggregates) and sizes of alginate nanogels were studied using scanning electron microscopy (SEM). Specimen preparation was performed as follows: the lyophilized nanogels were first suspended in ethanol, mounted on stubs and sputter-coated with gold. Micrographs were taken using an SEM instrument (Model S-4800 microscope, Hitachi, Tokyo, Japan). The nanogels mounted on stubs were further used to analyze the presence of metal cations by using an electron probe X-ray microanalyzer in energy dispersive X-ray spectrometry (EDX) mode.

#### 2.2.3. FT-IR Spectral Study

Samples of sodium alginate, urease, and encapsulated urease in different alginate nanogels cross-linked with Mn^2+^, Fe^3+^, and Co^2+^ were lyophilized before performing the FT-IR analysis. The FT-IR spectra were recorded on a Perkin-Elmer spectrometer (model: Spectrum RXI–Mid IR). The spectra were collected from 4000 to 400 cm^−^^1^ in the transmission mode.

### 2.3. Enzyme Assay

Enzyme activity of soluble and immobilized urease was determined by a spectrophotometric method [62]. The appropriate amount of soluble or immobilized urease was incubated in 0.1 M urea with intermittent shaking. The amount of NH_3_ liberated after incubation for a fixed time interval was determined using Nessler’s reagent. The absorbance was measured spectrophotometrically at 525 nm (Shimadzu UV-Vis spectrophotometer, Kyoto, Japan, UV-1800). One unit of urease activity liberated 1 μmol of NH_3_ from 0.1 M urea per min, under the standard assay conditions (0.05 M Tris-acetate buffer, pH 7.2, and 37 °C). The protein content in the alginate nanogel was determined by the method of de-cross-linking the alginate nanogels in brine solution, given by Pignolet et al. [63]. In this method, the breaking of cation–alginate cross-links releases urease protein, which was further estimated by Bradford’s method [64]. The residual protein content of the wash solution was also determined by the Bradford method of protein estimation [64]. The enzyme loading efficiencies and loading capacities of the nanogels were calculated as follows:A. Enzyme loading efficiency = (Wt. of urease in alginate sol-Wt. of residual urease)/(Wt. of urease in alginate sol) × 100(1)
B. Loading Capacity = (Wt. of urease in alginate sol-Wt. of residual urease)/(Polymer weight)(2)
(3)$$C.\;\mathrm{Percent}\;\mathrm{immobilization}\;\left(\%\right)=\frac{\mathrm{Specific}\;\mathrm{activity}\;\mathrm{of}\;\mathrm{immobilized}\;\mathrm{urease}}{\mathrm{Specific}\;\mathrm{activity}\;\mathrm{of}\;\mathrm{soluble}\;\mathrm{urease}}\times 100$$

The specific activity of immobilized urease was determined as detailed above. The specific activity of soluble urease was calculated by subtracting the specific activity of urease during washing (unbound urease) from the specific activity of total soluble enzyme.

### 2.4. Steady-State Kinetics

The effect of the substrate concentrations on urease activity was investigated at 25 °C by varying the urea concentration from 1 mM to 30 mM at optimum pH 7.2 for soluble and immobilized enzymes. The enzyme assay was performed as described earlier. *K_m_* and *V_max_* were determined from a Lineweaver–Burk plot and the turnover numbers (*K_cat_*) were calculated.

### 2.5. Storage and Stability Studies of Urease in Nanogels 

The soluble and immobilized urease were stored in 100 mM Tris-acetate-saline buffer at pH 7.2 and 4 °C. The activity was determined and recorded for four weeks at regular intervals for stored urease (immobilized and soluble) using the assay method (described in Section 2.3) under similar conditions. The percent residual activity was plotted against the number of days. The stability of immobilized urease enzyme in different nanogels with 90% residual enzyme activity was calculated in a Tris-acetate-saline buffer. The effect of substrate concentration on urease activity was investigated at 37 °C, by varying the urea concentration from 1 to 30 mM at optimum pH for soluble and immobilized enzyme.

## 3. Results and discussion

### 3.1. Alginate Nanogels by Emulsification Method

In the past two decades, tremendous efforts have been made towards the development of synthetic strategies for nanostructured materials with well-controlled size, shape, composition, and spatial arrangement [65,66]. Application of a pre-existing nanostructured template is one of the most effective strategies towards achieving this goal. Typically, nanomaterials synthesized by templating strategies hold a well-defined size, shape, and configuration, which usually benefits from the directing effect of the templates [66,67]. The application of microemulsion as a template is one of the most promising approaches for the synthesis of nanomaterials as they form a colloidal system with droplet sizes generally less than 100 nm [68,69,70]. The central idea of using thermodynamically stable surfactant templates (microemulsion) is to turn the dynamic molecular aggregates into a chemically and mechanically stable supramolecular material through templating reactions [71,72]. Water is usually considered as a necessary component of microemulsions; however, if the water in usual microemulsion recipes is replaced with a polar hydrophilic compound or polymer—such as sodium alginate sol (Na–alginate)— that is immiscible with the nonpolar phase, then microemulsions can also form. A substantial amount of literature is available about the nature of water in reverse micelles, formed from ionic surfactants such as AOT; however, significantly less is known about the nonionic reverse micelles with aqueous phase other than water [68,73].

Here, we chose the commonly employed W/O microemulsion of a nonionic surfactant for the preparation of alginate nanogels in an aqueous alginate (alginate sol)–hexane biphasic system. Hexane was chosen as oil phase because of its low dielectric constant and better optical contrast with the surfactant used in our study. Span 20 is a hydrophobic nonionic surfactant with a low hydrophilic–lipophilic balance (HLB = 8.6). It is a biodegradable surfactant that does not show much pronounced interaction with the polyvalent cations (Mn^2+^, Fe^3+^, and Co^2+^), as compared with the anionic and cationic surfactants, such as AOT (sodium bis[2-ethylhexyl] sulfosuccinate) or CTAB (cetyltri-methylammonium bromide). The critical micelle concentration (CMC) of lipophilic surfactant Span 20 in the hexane–water interface is reported in the literature [74,75], but the interfacial elasticities of Span 20 are influenced by the salt present in the aqueous phase [76]. Thus, for the development of microemulsion as template for alginate nanogel synthesis, it is desirable to form a stable emulsion/microemulsion of alginate sol–hexane bi-phasic system with a minimum concentration of surfactant. Hence, we applied DLS to study the microemulsion in the alginate sol–hexane system, which allows measurement of the reverse micelles’ hydrodynamic size and provides information on the surfactant’s CMC and formation of stable aggregation as template [77,78]. The results are shown in Figure 1. Each data point was averaged from three independent measurements, and the standard deviation was calculated. The hydrodynamic diameter of Span 20 aggregates in hexane initially increases with surfactant concentration and poly dispersity index (PDI) value (0.3–0.8), then remains roughly constant at around 110 nm (PDI value = 0.3–0.5) over a wide range of concentrations. The higher poly dispersity index of reverse micelles at the lower surfactant concentration suggests a larger variability in the size of aggregate formed by Span 20; additionally, the reverse micelles’ diameter become smaller with lower variability, as the increase in the surfactant concentration form stabilizes the aggregates [79]. 

The DLS results indicate a plateau value (reverse micelle size) above the CMC (10 mM), confirming spherical structures of reverse micelles. According to the literature, relatively larger microstructures are formed by Span 20-based surfactants [80], which is consistent with our DLS data shown in Figure 1. A value of about 110 nm for the hydrodynamic diameter for Span 20 reverse micelles in hexane can be inferred from the plateau in Figure 1. Alginate nanogel synthesis requires a stable template as nanoreactor for controlling the size of nanogels. Hence, at a fixed surfactant solution in hexane, a different volume of aqueous phase was dispersed by magnetic stirring (500 rpm) and the swollen reverse micelles or microemulsion droplets obtained in this way were studied with the DLS technique. As sodium alginate sol is insoluble in oil (hexane), the alginate polymer is confined within the aqueous nanophase, forming a droplet with defined dimensions. The phase stability and the droplet size of the resulting microemulsion is significantly affected by the amount of alginate sol added. The trial compositions were plotted on a ternary phase diagram, as shown in Figure 2.

At the constant Span 20 to hexane ratio, for a low alginate sol concentration, a visually transparent microemulsion region appeared, represented as square points in the phase diagram (Figure 2). The droplet size was 139 ± 3.43 (std) nm at point A in the ternary phase diagram for Span 20 (Figure 2); the corresponding microemulsion is shown in Figure 3 (A’). As the alginate aqueous phase concentration increases, the system becomes slightly turbid and translucent, see the circles in Figure 3, where point B’ represents a (micro) emulsion, with droplet size 311 ± 10 nm. At still higher alginate phase concentration, microemulsions with a milky appearance were obtained, as shown by the triangles in Figure 2, where, at point C’, the droplet size is 971 ± 5.25 nm. The corresponding microemulsions at points B’ and C’ are shown in Figure 3B’,C’, respectively.

For all systems studied, droplet size increases as the amount of aqueous phase increases. Thus, at the concentration of dispersed phase (alginate solution), showing droplet size 140 nm, the swollen reverse micelles were chosen as a template for the nanogel synthesis. In the present sol–gel process, based on microemulsion polymerization, the spherically swollen reverse micelles act as a water pool, enclosing the alginate sol [81,82]. Subsequent addition of divalent cation solutions (MnCl_2_, FeCl_3_, and CoCl_2_) induces gelation of these sol pools, with simultaneous phase separation and production of spherical gel particles containing Mn^2+^, Fe^3+^, or Co^2+^as cross-linkers, with a different hydrodynamic diameter than the original microemulsion droplets, as shown in Table 1. We used an alginate polymer with high mannuronate (M) content, because this composition is more stable for NaCl treatment [83]. Alginate nanogels, cross-linked with Mn^2+^, Fe^3+^, or Co^2+^ do not show any gradation in the hydrodynamic diameter. Although they have similar ionic radii, their affinity for mannuronate and guluronate are different. The hydrodynamic diameter of Mn–alginate nanogels are larger (275 nm) than the swollen reverse micelles (139 nm); whereas, Fe–alginate and Co–alginate shows smaller sizes. A simulation study in the literature [84,85] suggests that changes in the ion-binding mode of transition elements controls chain–chain association within junction zones, which could affect the size of corresponding nanogels. Interpretation from the literature [17,86,87,88] and quantum chemical calculations [23] suggest that 3d metallic ion forms different structural forms as a result of cooperative associations of several macromolecular chains of sodium alginate, which depends on the nature of the cation or the structure of the alginate chain. Co^2+^ show smaller cooperative interaction, and in Mn^2+^, cooperative interchain association is absent. These variations signify the altered internal morphology of metallic alginates, which could affect the size and shape of corresponding alginate nanogels.

### 3.2. Morphology Analysis by SEM& EDX Studies

The DLS-based studies provide ensemble averages of apparent particle radii. Further insight into the characters of colloidal nanogels morphology and dimensions were examined by studying the smears or films of colloidal nanogels in maximum swelling condition in Tris-buffer at pH 7.2 using SEM. The SEM micrographs of the nanogels are shown Figure 4A–C. 

The spherical alginate nanogels containing the polyvalent cations (Mn^2+^, Fe^3+^, or Co^2+^) have much smaller diameters than the hydrodynamic diameters measured by DLS and are relatively monodisperse. The particle size of alginate nanogels cross-linked with Mn^2+^, Fe^3+^, or Co^2+^ show no gradation, as observed in the DLS measurement studies. Mn–alginate and Co–alginate are spherical, whereas, Fe–alginate is elongated and rice-shaped under the same preparation condition. Thus, nanophase confinement of sodium alginate polymers during microemulsion polymerization does not control the gel size and shape. The interaction of ionic polysaccharides with gelling solution leads to the formation of junction zones that may affect the hydrodynamic size and shape of the nanogels. Studies of polyelectrolyte complexation in the various oil-in-water interfaces suggests that shape of particles can be easily tuned by the interfacial tension [37,89,90]. Thus, it can be concluded that the dimension of the template is not the key parameter in controlling the shape and size of nanomaterial [91]. The poly(lactide-co-glycolide) (PLGA) particles shape can be tailored by the balance between the polymer viscosity and the interfacial tension [92]. Similarly, different shapes of alginate particles cross-linked with Ca^2+^ ions can be obtained by changing the pH of the gelation solution, as it gives rise to a change in the interfacial tension [93]. A comparison of the cross-linking ability of Ca^2+^ and Ba^2+^ for gelation process by Chuang et al. suggests that Ba^2+^, acting as a cross-linker, had a less impact on the particle shape than Ca^2+^, due to a higher affinity in alginate intermolecular cross-linking [37]. In the present work, we used Tris-acetate-saline buffer (pH-7.2, 0.1 M) for the preparation of alginate sol and the development of the microemulsion for the encapsulation of protein in the nanogels cross-linked with Mn^2+^, Fe^3+^, or Co^2+^ ions, under physiologically compatible pH conditions. The pH of the gelling solutions of the MnCl_2_ (pH = 5.20) and CoCl_2_ (pH = 5.30) salts were similar, and they form spherical particle of different sizes. In the case of Fe–alginate synthesis, the gelling solution of FeCl_3_ has pH 2.46, which could affect the interfacial tension of alginate sol/hexane/Span 20 microemulsion, resulting into the anisotropic shape (rice shaped) of Fe–alginate nanogels. The results corroborate the observations reported by Chuang et al. [37].

We have used a 0.1 M gelling solution to replace the sodium ions from the linear polymeric chain of sodium alginate. Thus, to confirm the replacement of sodium ions by divalent gelling ions, EDX spectra were recorded as shown in Figure 4A’–C’. The spectra shows the relative proportion of elements present in the different nanogels and contain cation peaks for the replacement of polyvalent cations (Mn^2+^, Fe^3+^, or Co^2+^). A small characteristic peak for sodium ion was also observed in the spectra, suggesting partial removal of Na^+^ by polyvalent cations, which forms 3D junction zones in the alginate nanogel structure.

### 3.3. Characterization of Alginate Nanogels Cross-Linked with Polyvalent Cations (Mn, Fe, or Co) by FT-IR Studies

Sodium alginate (powder), lyophilized alginate nanogels containing different cations, and urease-encapsulated nanogels were analyzed using an FT-IR spectrophotometer to study cation–alginate interactions before and after gelation and encapsulation of proteins. The spectra of the cross-linked alginate nanogels are quite similar. The corresponding FT-IR spectra are given in the Appendix A. There are four particularly relevant spectral bands in the alginate nanogels prepared and tested under the same conditions as those shown in Table 2A. The ν(O-H) (1) bands are broadened and shifted to lower wave numbers in nanogel structure cross-linked with polyvalent cation (Mn^2+^, Fe^3+^, or Co^2+^), compared with linear polymeric Na–alginates, indicating that the O–H bond is weakened due to hydrogen bonding in the gel structure [94]. The ratio of intensities of ν(C=O) (2) and ν(C-OH) (3) suggests the presence of a protonated carboxylic group in the nanogels [95]. Band (4) indicates the presence of an O-glycosidic bond between β-d-mannuronic and α-l-guluronic acid residues in the linear alginate chain [96]. The bands in these regions are broadened and smoothed with a shift to lower wave number relative to sodium alginate in alginate nanogels formed by Mn^2+^, Fe^3 +^, and Co^2+^, and do not show much deviation. This suggests that the O-glycosidic bonds between β-d-mannuronic and α-l-guluronic acid residues in the metallic alginate gels are not perturbed due to gel formation.

The FT-IR spectra of all the urease-encapsulated cross-linked alginates are shown in Table 2B. They are quite similar to corresponding metal alginates showing ν(O-H) (1) bands, symmetric and asymmetric stretching bands of carboxylic acid (ν_sym_(COO^−^) (2), ν_asym_(COO^−^) (3)) given in Table 2A, but they show a small shift in their positions. These observable changes could be due to the overlapping with the characteristic vibrational bands of encapsulated proteins [97]. In the spectra of the urease-bound metal alginates, a significant new peak (5) appears at about 1250 cm^−1^, due to C–N stretching vibrations, which suggest proper encapsulation of urease in the alginate matrix [98]. The characteristic amide band for protein are non-distinguishable, as it is reported that these infrared bands are broad and often overlap with neighboring bands to produce a complex absorption profile [99]. The corresponding FT-IR spectra are given in the Appendix A.

### 3.4. Surface Charge and Stability

The interaction of a protein with polymers and the other biomaterials used in biomedical applications has characteristic electrical properties, such as the local electrostatic charge distribution and the electrical double layer potential, which play significant roles in defining the biological interactions, aggregation behavior, and stability [100]. The zeta potential (ZP) is an indicator of the surface charge properties of a colloid or a particle in solution and depends on the surface potential and the thickness of the electric double layer. The zeta potentials of nanoparticles with charged functional groups at the surface are measured to determine their colloidal stability by coulombic repulsion [101], which is important for their applications. The alginate sol present in the microemulsion shows highest zeta potential (−57.13 ± 0.33) confirming the stability of the microemulsion templates by coulombic repulsion and the presence of a negatively-charged carboxylic functional group in the aqueous droplets. The alginate nanogels were formed by electrostatic interaction between the negatively charged carboxylic groups and hydroxyl groups of alginate polymer chain and the positively charged polyvalent cations (Mn^2+^, Fe^3+^, and Co^2+^) that form a cross-linked network containing a large fraction of water in the microstructure. Thus, the negative charge decreases during the gelation of alginate sol. The negative zeta potential values shown in Table 3 indicate an open and porous gel network with free carboxylic groups at the surfaces of the alginate nanogels, which cause electrostatic repulsion among the nanogel structures [102,103]. Among all the alginate nanogels studied, Fe–alginate nanogel has highest value, which suggests that it has a more open gel network and coulombic stability than the other cross-linked alginate nanogels. The corresponding zeta potential graphs are given in the Appendix A.

### 3.5. Enzyme Assay, Using Urease Enzyme Encapsulated in Alginate Nanogels

The immobilization technique combines the advantage of microemulsion technology to form alginate-based nano-droplets and alginate’s gelling property to encapsulate urease providing them a gentle hydrated and nontoxic environment with higher enzyme activity retention. The efficacy of enzyme immobilization technique was studied by evaluating various immobilization parameters, enzyme kinetics, reusability, storage, and stability studies in the alginate nanogels cross-linked with different polyvalent cations (Mn^2+^, Fe^3+^, and Co^2+^) to choose the best matrix for the immobilization of proteins. Literature studies [104,105,106] suggest that size of alginate particles formed by gelation method increases with the increase in the concentration of alginate sol. In the present study, we have determined the effect of alginate solutions on the development of swollen revere micelles, and it was found that 0.2% alginate sol (*w/v*) is suitable for the development of urease-encapsulated alginate nanogels cross-linked by different cations (Mn^2+^, Fe^3+^, or Co^2+^) with the size of about 100 nm. Hence, for the immobilization of urease in alginate nanogels, different concentration of lyophilized urease was pre-mixed with 0.2% alginate sol to form urease-encapsulated alginate nanogels. During encapsulation of urease by the sol–gel transformation, a fraction of protein is immobilized in the 3D structure of alginate nanogels, while the rest remain unbound. Enzyme loading efficiency and percent immobilization are determined to describe efficacy of enzyme immobilization inside a solid support as it gives insight about the activity of immobilized enzyme [55]. By comparing these parameters, we can predict the activity per unit mass of bound protein in the solid support. A comparison of percent immobilization and enzyme loading efficiency (Table 4) suggests that, as we increase the concentration of protein in the premix alginate sol from 1 mg per ml to 5 mg per ml, percent immobilization value (i.e., the specific activity of immobilized enzyme) increases, while a further increase in the protein concentration (7 mg/mL) has an adverse effect on percent immobilization. Percent immobilization represents the fraction of biocatalyst activity present in the solid support, in comparison to the specific activity of total soluble enzyme used in the sol–gel transformation process [62]. As concentration of protein in premix sol increases, over-crowding of protein molecule in the nanogel microenvironment may hinder the internal diffusion of the reactant or product and, thus, the enzyme activity decreases. The immobilized enzyme expresses only a fraction of the expected activity due to enzyme inactivation, steric hindrances, or mass-transfer limitations, while the unbound enzyme may become inactive later [107,108,109].

Enzyme loading efficiency is simply the ratio of bound protein inside the alginate nanogel structure and the total protein used in the pre-gel mixture. In cases of nanogels cross-linked with 3d metallic ions (Mn^2+^, Fe^3+^, and Co^2+^), the percent immobilization values are similar to, or lower than, the enzyme loading efficiency data (Table 4), suggesting the partitioned microenvironment of alginate nanogel structure had no effect. But the mass transfer limitations and steric hindrances could be relevant and pronounced in these nanogel structures, which affects the enzyme activity by retarding the rate of the biochemical reaction. Similar effects have been reported in many enzyme-immobilized matrices [108,109,110,111], which supports our finding.

#### 3.5.1. pH Based Stability Studies

Enzymes are zwitter ionic in nature, in these, the charge distribution property changes with the pH of the bulk phase. The rate of enzymatic hydrolysis of urea by immobilized urease enzyme changes with pH of the medium. Thus, it is important to study the effect of pH on the immobilized urease enzyme activity. We have measured the activity of encapsulated urease in alginate nanogels cross-linked with Mn^2+^, Fe^3+^, and Co^2+^, as shown in Figure 5, within the maximum urease activity range (pH 5–10), with 10 mM urea solution at 37 ± 1 °C. The maximum urease activity (percent) measured by the enzyme assay method (Section 2.3) was obtained at pH 7.2, as shown in Figure 5. Additionally, it is close to the value reported in MSDS from the Sigma chemical Co. (i.e., 7.4), as they have used Tris-HCl to measure pH optima. Howell, S.F., Sumner, and J.B. [112] have also studied the effect of different buffers on urease activity and concluded that activity of enzyme depends on the type of buffer, temperature, and salt concentration. Similarly, Illanes**,** Andrés, and others [111,113] have also reported the effects of matrix on pH maxima of different enzyme. Enzymes are poly ionic in nature and the charge property of proteins are governed by type of buffer, salt, and temperature as it affects the charge distribution and protein structure. In the present study, we studied the pH effect of a Tris-acetate-saline buffer for urease activity measurement. There was no effect of different metal ions cross-linked in alginate gels on the pH optima; thus, in the present study, pH-based microenvironments were important for the functioning of enzyme active site leading to biochemical transformations. 

#### 3.5.2. Steady-State Kinetics 

Measurements of the biocatalytic properties of urease immobilized on a solid support (i.e., alginate nanogels) forms a heterogeneous system where enzyme is in one phase, and the substrate or product is present in bulk aqueous phase, with a biochemical reaction taking place at the surface or inside the nanoparticle (immobilized biocatalyst). Thus, substrate or product diffuses through the nanogel pores which may affect the mass transport process. The enzymatic hydrolysis of urea can be represented by the following overall reaction [114]:(4)$${\left({\mathrm{NH}}_{2}\right)}_{2}\mathrm{CO}\;+2H_{2}O\;\to 2{\mathrm{NH}}_{4}^{+}+{\;\mathrm{CO}}_{3}^{2-}$$

Assuming a noncompetitive mechanism for ammonium detection, using Nessler’s reagent [115] leads to the following rate expression:(5)$$r=V_{max}\cdot \left[S\right]/\left(K_{M}+\left[S\right]\right)\left(1+\left[P\right]/K_{P}\right)$$
where [*S*] and [*P*] are the substrate and ammonium ion concentrations, respectively; *V_max_* is the maximum reaction rate; *K**_m_* is the Michaelis–Menten constant; and *K**_P_* is the dissociation constant for the enzyme–product complex.

Kinetic data were analyzed by the initial rate method, and (*K**_m_*) and *V**_max_* values were obtained from Lineweaver–Burk plots for the different nanogels, as shown in Table 5. The range of concentration of urea studied was 1–30 mM (~6 mg/dL to 180 mg/dL), which can be applied for measurements of blood serum urea in normal, as well as kidney, patients, and milk–urea can be applied as food adulterant. The calibration curves were obtained for alginates gels synthesized using MnCl_2,_ FeCl_3,_ and CoCl_2_ gelling solutions.

The Lineweaver–Burk plot was obtained at the enzyme concentration of 5 mg/mL at which, maximum percent immobilization was obtained for all the different nanogels studied (as give in Appendix A). The nature of the double reciprocal plot is an indicator of enzyme–substrate interaction attributes in a heterogeneous system, where the enzyme is immobilized inside a solid support and substrates or products present in a bulk solution undergo mass transport. The various parameters of mass transport and steric hindrance are difficult to calculate. But the nature of plot is showing non-linear behavior with Mn–alginate, which suggests that internal diffusion restrictions and steric hindrance is possible in enzyme–substrate interactions inside the gel matrix. But the linear nature of double–reciprocal plot with Fe–alginate and Co–alginate suggests that carrier matrix is compatible and expresses maximum activity with urea [111]. The *K_m_* value obtained from the linear plot is smaller than the *K_m_* value reported for soluble urease in all the cases except for Mn–alginate, which indicates that entrapment of urease inside the compartmentalized, porous nanogels is favorable [109,116]. 

#### 3.5.3. Storage Stability of Urease in Nanogel

All forms of urease were stored at 4 °C in 100 mM Tris-acetate-saline buffers at pH 7.2. The stabilities of the different urease preparations were determined by measuring the specific enzyme activity of immobilized urease and soluble urease in Tris-acetate-saline buffer at 25 ± 1 °C, using the urease assay method over a period of four weeks. The percent (%) of residual activity, versus the number of days, is shown in Figure 6. Enzyme activity retention depends on the nature of the carrier and the nature of the enzyme [116]. The zeta potential of alginate nanogels, reported in Table 3, has negative value and these suggest the open porous structure of different nanogels. The pH value of urease is 5.97, as reported in the literature [117]. Thus, at pH 7.2, the carboxylic function of the alginate gel structure shows repulsion towards the negatively charged urease. With time, these carboxylic functional group present on alginate matrix may change the microenvironment of porous nanogels, leading to the loss of enzymes [111]. The encapsulated urease enzyme in Fe–alginate nanogels showed maximum stability (90% activity), up to 6 days. The zeta potential value, which is a measure of stability of nanogels in water, is also largest for the Fe–alginate nanogels. Thus, colloidal stability helps in the retention of enzyme activity in Fe–alginate nanogels.

### 3.6. Analytical Application of the Fe–Alginate Nanogels

The colloidal stability and enzyme activity retention of Fe–alginate is maximum among all the nanogels synthesized in this study. It was applied to determine the urea concentrations in anonymized blood serum samples obtained from the hospital. The measurements were performed using standard addition methods in triplicate. To obtain stable readings, the conditions for measurements (amount of particle, total test volume, and incubation time) were established for a urea concentration of 180 mg/dL (30 mM) as test analyte—which can be applied for the measurement of blood serum urea in normal humans as well as kidney patients. This is much higher than the normal physiological range.

The results obtained with the present enzyme assay protocol and with the clinical method [118] are compared in Table 6. The urea content obtained with the present system was always higher than that measured by the clinical method, probably due to interference from other compounds present in blood serum. In healthy subjects, blood urea levels are typically in the range 7–20 mg/dL [118], which increases above 50 mg/dL in kidney failure [119]. The present study shows that our method was satisfactorily applicable to samples containing 5–25 mM (90.090–450.450 mg/dL) of blood glucose, with a relative error of not more than 5%. Thus, the urease-encapsulated, Fe–alginate nanogel-based measurement can identify a kidney patient.

## 4. Conclusions

Alginate nanogels of different sizes were successfully prepared using a mild method that exploits surfactant self-assembly, as a template for converting molecular aggregates into stable nanogels by ionotropic sol–gel transformation. In this work, we focused on the basic properties of Span 20 surfactant, which forms stable, spherical reverse micellar-type aggregates of well-defined size at 110 nm, and at a surfactant concentration above CMC (10 mM). DLS measurement studies of different alginate nanogels indicate that the high M–alginate nanogels obtained in the presence of Mn^2+^, Fe^3+^, and Co^2+^ showed no gradation in the hydrodynamic diameter. Urease enzyme was successfully encapsulated in these alginate nanogels. A comparative assay of urease enzyme demonstrates that Fe–alginate has 90% urease encapsulation efficiency, with the highest turnover number and linear range of urea detection. Thus, bioactivity was best retained in the protein-loaded Fe–alginate nanogels with anisotropic shape (rice shaped). The Fe–alginate nanogel was able to protect and preserve enzyme activity and stability during particle formulation, recovery, and storage in 100 mM Tris-acetate-saline buffer. The present study demonstrates that the shape and size of the self-assembly template of Span 20 in alginate sol/hexane microemulsion do not control the size and shape of nanogels formed with 3d transition metal ions (Mn^2+^, Fe^3+^, and Co^2+^) as cross-linkers. The elongated, rice-shape of the Fe–alginate has the influences on its morphology, which, in turn, affects both the loading capacity and the catalytic activity of urease enzyme. The clinical study revealed that immobilized urease in Fe–alginate nanogels can be used measure urea in blood serum samples. The nanogels developed in this work are viewed as promising candidates for biosensing, drug delivery, and many other applications, such as cellular targeting and uptake related to shape-specific biological behaviors [120].

The structural elucidation of the alginate gels with cross linking to the cations have emerged as part of vital data that can correlate the modulations at the level of stereochemical and cross-linking architecture. These can further be utilized to investigate the possible simulations of biological perturbations for demarcating effects of metal-bound alginates. The studies conducted here have promising implications in biomedical research indicating the crucial role played by the structural affiliations of the synthesized alginates with bioactive enzymes, that can be extended to explore their applications within the nano-framework.

## 5. Patents

The present research study is a part of our provisional patent application (Application No. 202111045884, Indian Patent office). 

## Figures and Tables

**Figure 1 polymers-14-01277-f001:**
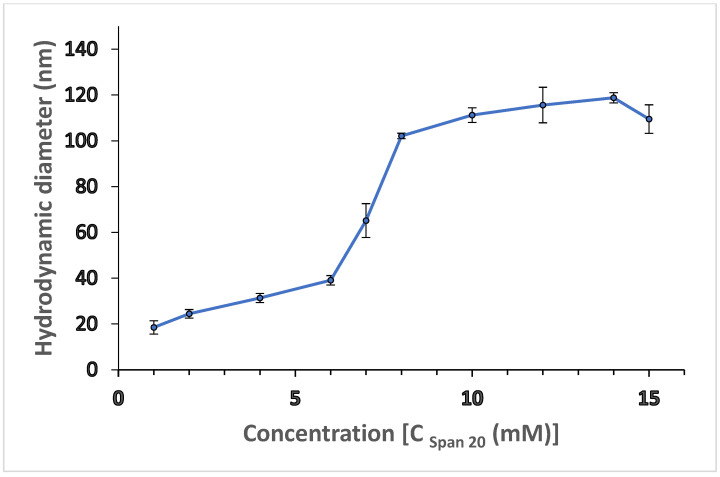
Apparent hydrodynamic diameter of aggregates formed by SPAN 20 in hexane, measured by DLS. Error bars are standard deviations.

**Figure 2 polymers-14-01277-f002:**
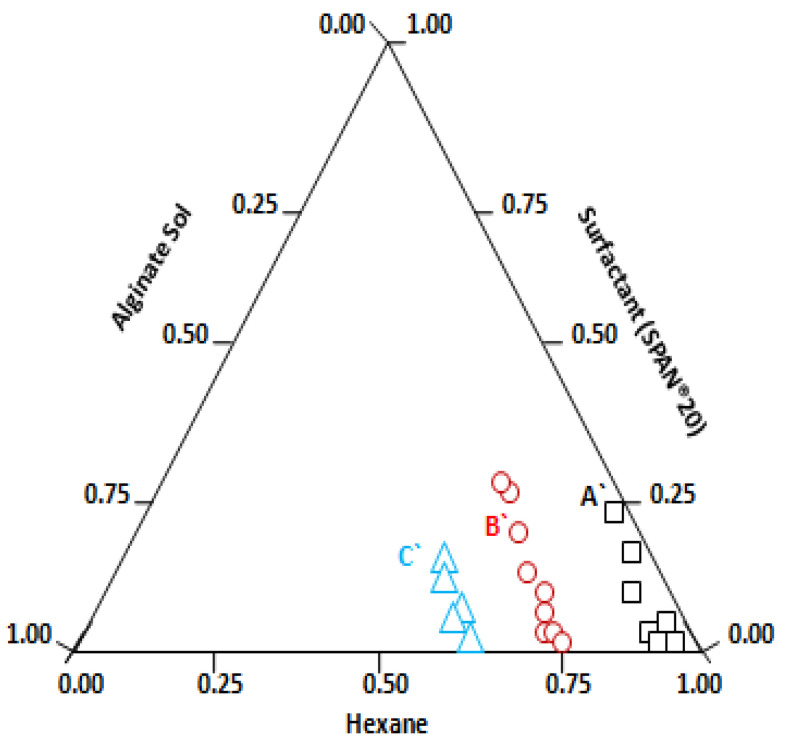
Phase diagram for alginate sol/hexane/Span 20 microemulsions. Squares (
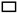
), circles (
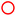
) and triangles (
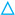
) correspond to the microemulsion regions where the micelle size varies from 110 nm to 231 nm, 431 nm to 586 nm, and 886 nm to 1163 nm, respectively.

**Figure 3 polymers-14-01277-f003:**
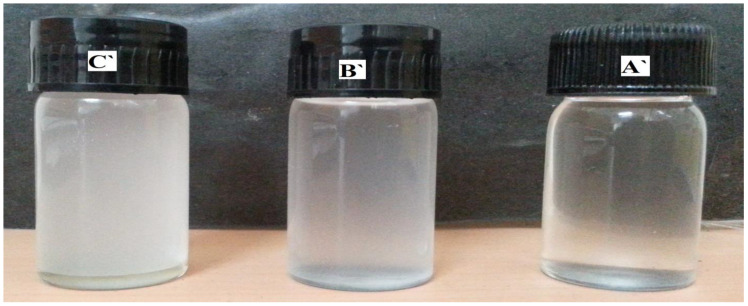
W/O microemulsion of Span 20, representing points (**A’**) (clear microemulsion), (**B’**) (slightly turbid microemulsion) and (**C’**) (milky appearance microemulsion).

**Figure 4 polymers-14-01277-f004:**
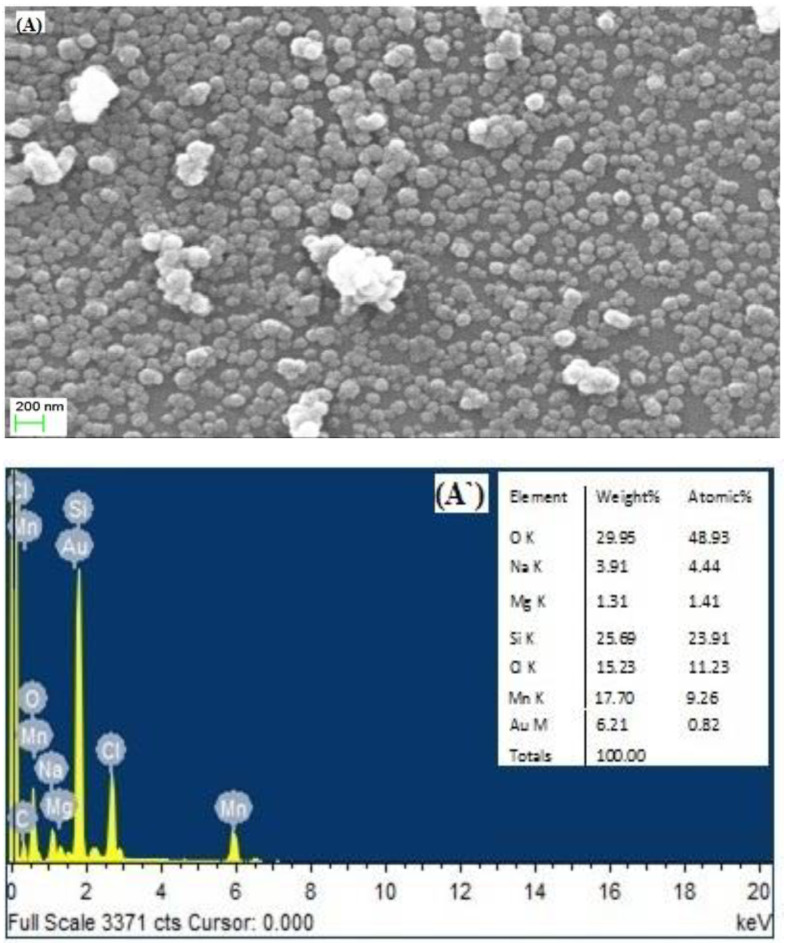
SEM (**A–C**) and EDX (**A’–C’**) micrograph of Mn–alginate, Fe–alginate, Co–alginate nanogels.

**Figure 5 polymers-14-01277-f005:**
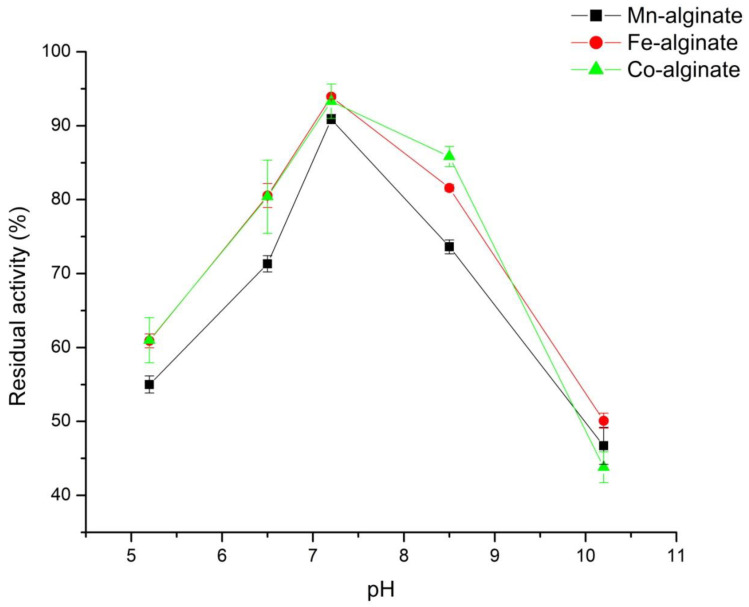
Variation of percent-relative activity with pH, studied for encapsulated urease enzyme in alginate nanogels cross-linked with Mn^2+^, Fe^3+^, and Co^2+^.

**Figure 6 polymers-14-01277-f006:**
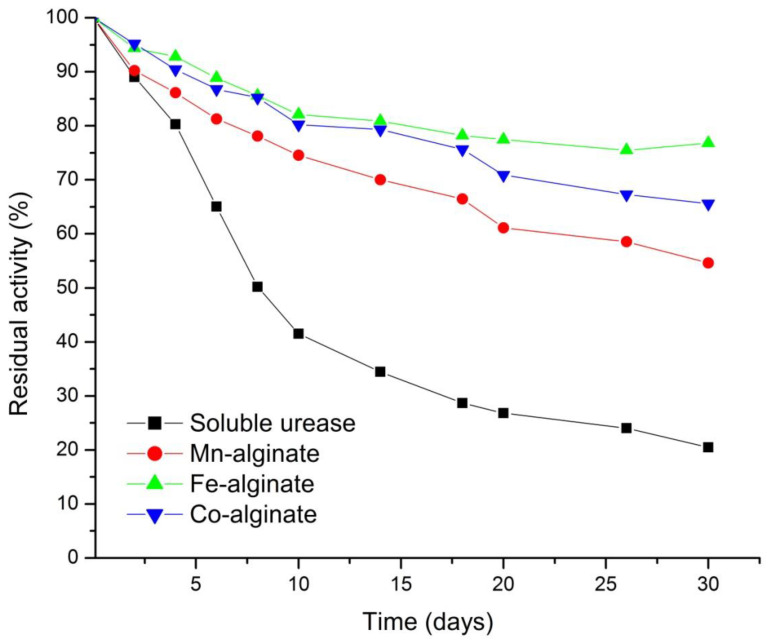
Variation of percent residual activity of immobilized enzyme with time for alginate nanogels cross-linked with Mn^2+^, Fe^3+^, and Co^2+^.

**Table 1 polymers-14-01277-t001:** DLS data obtained from microemulsion with different divalent cation solutions collected before, and 1 h after, the addition of divalent cation solutions.

S. No.	Gelling Solution (0.1 M–2 mL)	Droplet Size of Microemulsion without Aqueous Phase (nm) before Gelation	Recovered Nanogels with Cations (nm) after Gelation	Shape	Cationic Size (Å)	Number of Cross-Linking Ions (ICPMS) (PPM)
1.	MnCl_2_	110 ± 1.42	275 ± 3.12	Spherical	0.67	197
2.	FeCl_3_	110 ± 1.42	120 ± 5.77	Spherical	0.61	248
3.	CoCl_2_	110 ± 1.42	7.6 ± 3.38	Spherical	0.65	186

Note: The DLS measurement data of microemulsion droplets and nanogels are given in Appendix A.

**Table 2 polymers-14-01277-t002:** (**A**) FT-IR spectral bands of alginate salt and nanogels cross-linked with 3d metallic ions (Mn^2+^, Fe^3+^, and Co^2+^). (**B**) FT-IR spectral bands of urease (lyophilized) and urease-encapsulated alginate nanogels formed by Mn^2+^, Fe^3+^, and Co^2+^.

**(A)**				
**Spectral Band**	**Na–Alginate Salt (Powder)**	**Mn–Alginate Lyophilized Nanogel**	**Fe–Alginate Lyophilized Nanogel**	**Co–Alginate Lyophilized Nanogel**
**ν(O–H) (1)**	3496	3472	3482	3512
**ν(C = O) (2)**	1646	1644	1648	1672
**ν(C–OH) (3)**	1474	1447	1464	1464
**ν(OC–OH) (4)**	1110	1116	1118	1134
**(B)**				
**Spectral Band**	**Urease (Free)**	**Urease-Encapsulated Mn–Alginate**	**Urease-Encapsulated Fe–Alginate**	**Urease-Encapsulated Co–Alginate**
**ν(O–H) (1)**	3412	3420	3382	3448
**ν_sym_(COO^−^) (2)**	1610	1640	1599	1652
**ν_asym_(COO^−^) (3)**	---	1439	1404	1452
**ν(OC–OH) (4)**	1458	1110	1074	1116

**Table 3 polymers-14-01277-t003:** Zeta potentials of linear sodium alginate chain and alginate nanogels cross-linked with divalent cations (Mn^2+^, Fe^3+^, and Co^2+^).

S. No.	Type of Alginates	Zeta Potential (mV)
1.	Sodium alginate (microemulsion droplet)	−57.13 ± 0.33
2.	Mn–alginate nanogels	−2.63 ± 0.02
3.	Fe–alginate nanogels	−8.78 ± 0.02
4.	Co–alginate nanogels	−4.83 ± 0.02

**Table 4 polymers-14-01277-t004:** Parameters of enzyme immobilization measured in alginate nanogels cross-linked with polyvalent cations (Mn^2+^, Fe^3+^, or Co^2+^).

S. No.	Type of Alginate Nanogel Containing Urease	Concentration of Sodium Alginate % (*w/v*)	Protein/mL of Alginate Sol (mg)	% Immobilization	Enzyme Loading Efficiency (%)	Enzyme Loading Capacity (10^−3^)
1.	Urease	0.0	-	-	100	-
2.	Mn–alg	0.2	1	52.7	68.4	0.6793
		0.2	5	58.0	52.5	0.5214
		0.2	7	35.14	70.16	0.6968
3.	Fe–alg	0.2	1	47.72	63	0.6257
		0.2	5	75.3	76.9	0.7637
		0.2	7	38.41	81.25	0.8069
4.	Co–alg	0.2	1	48.32	67.2	0.6674
		0.2	5	59.7	66.4	0.6594
		0.2	7	35.17	80.8	0.8025

Note: All the readings are an average of six similar experiments.

**Table 5 polymers-14-01277-t005:** Kinetic parameters of immobilized urease enzyme in different alginate nanogels cross-linked with Mn^2+^, Fe^3+^, and Co^2+^.

S. No.	Type of Alginate Nanogel Containing Urease	*V_max_* (mmol/min)	*K_m_* (mM)	Linear Range of Calibration Curve (mM)	Turn Over No.
1.	Urease	--	2.4	--	--
2.	Mn–alg	1.16	2.597	5.0–15.0	49.42
3.	Fe–alg	1.62	0.31	5.0–25.0	80.52
4.	Co–alg	1.33	0.51	1.0–5.0	61.29

Note: All the readings are an average of six similar experiments.

**Table 6 polymers-14-01277-t006:** Determination of urea in blood serum samples using the urease-encapsulated Fe–alginate nanogels and a comparative spectrophotometric method used in clinical laboratories.

S. No.	Urea Concentration (mg/dL) (Clinical Method) $\upsilon_{E}$	Urea Concentration (mg/dL) (Present Method) $\upsilon_{A}$	Relative Error $\left[\;\frac{\upsilon_{A}-\upsilon_{E}}{\upsilon_{E}}\;\times 100\;\right]$
1.	35.8	36.4 ± 2.5	1.675
2.	50.4	52.4 ± 4.5	3.968
3.	82.5	85.1 ± 2.2	3.151
4.	116.4	119.6 ± 2.6	2.749
5.	129.8	135.4 ± 2.4	4.314
6.	138.4	145.1 ± 2.5	4.841
7	143	150.2 ± 3	5.034

## Data Availability

The data presented in this study are available on request from the corresponding author.

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
