# Peer review of "Synthesis of Alginate Nanogels with Polyvalent 3D Transition Metal Cations: Applications in Urease Immobilization"

_polymers, 2022, doi:10.3390/polym14071277_

Round 1

Reviewer 1 Report

Comments

  1. Be consistent throughout the manuscript in mentioning the term “microemulsion”. At some places it was mentioned as “emulsion” as shown in Table 1.
  2. As the developed preparation is “nanogels”, therefore, rheological study for its characterization is also important parameter, but it is missing in the present work.
  3. Gelling strength of the developed nanogels should also be measured.
  4. There are so many typo errors through the manuscript, at some places two words or more than two words are in joint form, e.g. “preeminentimportance”, in abstract “110 nmin (Line 14)”, studywe (Line 11) etc. and so on. There should be a space between the word and (REFERENCE), correction needed through the manuscript. Similarly, after full stop, there should be space before starting the next sentence (e.g. Line 66, Line 69, Line 102 and so on).
  5. Correct the caption “Figure 3. W/O microemulsion of Span20 representing points A` (clear solution-microemulsion), B` (slightly turbid solution) and C` (milky appearance-emulsion)” either it can be microemulsion or solution it cannot be solution-microemulsion.
  6. In line 51: As mentioned “Varied metal cations (Cu2+, Zn2+, Sr2+, …)” remove the dots and put etc.
  7. The source of procured materials as mentioned in “Materials and methods” should be presented properly, like Company, City, and Country. If equipments then Model also.
  8. Same instrument was used for size, PDI and zeta potential measurements, so, sub-section 2.2.1 and 2.2.4 should be merged and put together after reducing number of word counts.
  9. Correction needed (not readable properly due to conjunction of words) in the presentation of equations (A) and (B).
  10. The representation of degree centigrade “0C” is different at different places in the manuscript as it can be seen in section 2.6. in this section authors mentioned “Figure 6”, where are the Figures 1-5, therefore maintain the sequence in presenting the Figure numbers.
  11. Line 224-243: what is the difference between section: 2.6. Storage and stability studies of urease in nanogels and section: 2.6. Stability studies subsection: 2.6.1. Storage stability, it is very confusing, needs correction.
  12. Polydispersity index determination was mentioned in the method, but no results about PDI and its discussion has been included in the manuscript. It is an important parameter to see the width of distribution and also the stability of any colloidal system.

Author Response

Dear Reviewer,

We appreciate your continuous support for the revision of this manuscript. We have tried to resolve all the queries raised by you through the e-mail dated 04-October -2021. We believe that it has resulted in an upgraded, revised manuscript, which you will find uploaded alongside this document. The revisions are highlighted with green text highlight color. The reviewer’s comments and their responses are also uploaded alongside this letter for your perusal.

Reviewer 2 Report

In this work, the authors developed the water-in-oil emulsion of sodium alginate-sol/hexane/span 20 as a template for controlled synthesis of alginate nanogels crosslinked with 3d-transition metal cations. This work looked like a continuation of the authors’ previous work (Ref. 55: J. Colloid Interface Sci.2017, 490, 452-461). The differences were that 3d transition metal ion (Mn+2, Fe+3 and Co+2) were used instead of Ca+2, Sr+2, and Ba+2 as cross-linker, span 20 instead of Tween 80 as surfactant. Compared with their published work, the authors should point out the innovation of this article.

Many spaces between words in the article were missing, which increased the difficulty of reading.

The as-prepared nanogels for clinical analysis of urea in blood serum samples should be added.

This paper needs major revision before it is suitable for publication in Polymers.

Author Response

Dear Reviewer,

We appreciate your continuous support for the revision of this manuscript. We have tried to resolve all the queries raised by you through the e-mail dated 04-October -2021. We believe that it has resulted in an upgraded, revised manuscript, which you will find uploaded alongside this document. The revisions are highlighted with green text highlight color. The reviewer’s comments are also uploaded alongside this letter for your perusal.

Round 2

Reviewer 1 Report

Authors have revised the manuscript as per the raised comments.

Reviewer 2 Report

The authors had carefully revised the manuscript. My concerns and questions had been addressed. Its publication in Polymers is recommended.